# Hi3D: Pursuing High-Resolution Image-to-3D Generation with Video Diffusion Models

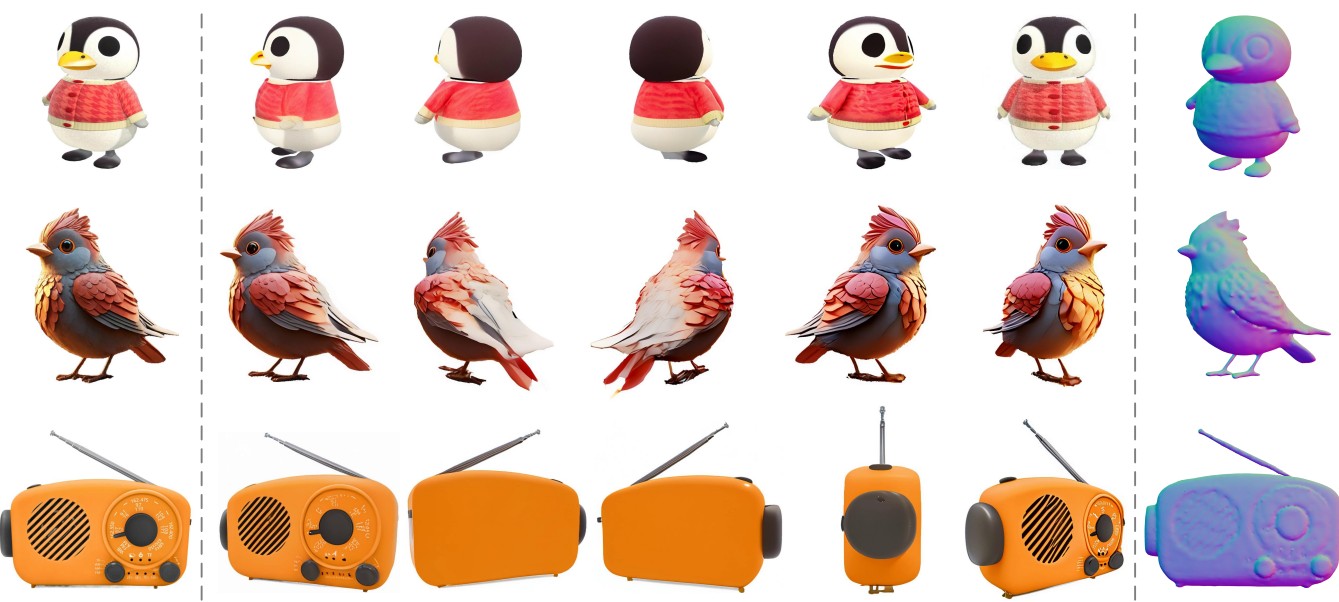

| Input image | Generated multi-view consistent and high-resolution sequential images | Mesh |

**Figure 1: Hi3D is capable of generating multi-view consistent and high-resolution (1,024×1,024) sequential images from an input single-view image of any style (top to bottom: images created by artists, generated by AI, or captured from the real world). Subsequently, we manage to reconstruct a high-fidelity 3D mesh conditioned on such high-resolution multi-view images.**

## ABSTRACT

Despite having tremendous progress in image-to-3D generation, existing methods still struggle to produce multi-view consistent images with high-resolution textures in detail, especially in the paradigm of 2D diffusion that lacks 3D awareness. In this work, we present High-resolution Image-to-3D model (Hi3D), a new video diffusion based paradigm that redefines a single image to multi-view images as 3D-aware sequential image generation (i.e., orbital video generation). This methodology delves into the underlying temporal consistency knowledge in video diffusion model that generalizes well to geometry consistency across multiple views in 3D generation. Technically, Hi3D first empowers the pre-trained video diffusion model with 3D-aware prior (camera pose condition), yielding multi-view images with low-resolution texture details. A 3D-aware video-to-video refiner is learnt to further scale up the multi-view images with high-resolution texture details. Such high-resolution multi-view images are further augmented with novel views through 3D Gaussian Splatting, which are finally leveraged to obtain high-fidelity meshes via 3D reconstruction. Extensive experiments on both novel view synthesis and single view reconstruction demonstrate that our Hi3D manages to produce superior multi-view consistency images with highly-detailed textures.

## KEYWORDS

Image-to-3D generation, Diffusion model, 3D generation

## 1 INTRODUCTION

Image-to-3D generation, i.e., the task of reconstructing 3D mesh of object with corresponding texture from only a single-view image, has been a fundamental problem in multimedia and computer vision fields for decades. In the early stage, the typical solution is to capitalize on regression or retrieval approaches [27, 57] for 3D reconstruction, which tend to be confined to close-world data with category-specific priors. This direction inevitably fails to scale up in real-world data. Recently, the success of diffusion models [18, 19] has led to widespread dominance for open-world image content creation [40, 48–50]. Inspired by this, modern image-to-3D studies turn the focus on exploring how to exploit 2D prior knowledge from the pre-trained 2D diffusion model for image-to-3D generation in a two-phase manner, i.e., first multi-view images generation and then 3D

reconstruction. One representative practice Zero123 [29] remoulds the text-to-image 2D diffusion model for viewpoint-conditioned image translation, which exhibits promising zero-shot generalization capability for novel view synthesis. Nevertheless, such independent modeling between the input single-view image and each novel-view image might result in severe geometry inconsistency across multiple views. To alleviate this issue, several subsequent works [21, 30, 32, 51, 52, 55] further upgrade the 2D diffusion paradigm by simultaneously triggering image translation between the input single-view image and multi-view images. Despite improving multi-view images generation, these approaches in 2D diffusion paradigm still suffer from multi-view inconsistency issues especially for complex object geometry. The underlying rationale is that the pre-trained 2D diffusion model is exclusively trained on individual 2D images, therefore lacking 3D awareness and resulting in sub-optimal multi-view consistency. Moreover, the geometry inconsistency among the output multi-view images will affect the overall stability of single-to-multi-view image translation during training. Hence, existing Image-to-3D techniques [21, 30, 32] mostly reduce the image size to low resolution (256×256). Such way practically increases batch size and improves training stability, while sacrificing the visual quality of output image of each view. This severely hinders their applicability in many real-world scenarios that require high-fidelity 3D mesh with higher-resolution texture details, such as Virtual Reality and 3D film production.

In response to the above issues, our work paves a new way to formulate image translation across different views as 3D-aware sequential image generation (i.e., orbital video generation) by capitalizing on the pre-trained video diffusion model. Different from 2D diffusion model that lacks 3D awareness, video diffusion model is trained with a large volume of sequential frame images, and the learnt temporal consistency knowledge among frames can be naturally interpreted as one kind of 3D geometry consistency across multi-view images, especially for orbital videos. This motivates us to excavate such 3D prior knowledge from the pre-trained video diffusion model to enhance image-to-3D generation. More importantly, such video diffusion based paradigm enables more stable sequential image generation with amplified 3D geometry consistency. It in turn allows flexible scaling up of higher-resolution sequential image generation (e.g., 256×256 → 1,024×1,024), triggering 3D mesh generation with higher-resolution texture details.

By consolidating the idea of framing image-to-3D in video diffusion based paradigm, we novelly present High-resolution Image-to-3D model (Hi3D), to facilitate the generation of multi-view consistent meshes with high-resolution detailed textures in two-stage manner. Specifically, in the first stage, a pre-trained video diffusion model is remoulded with additional condition of camera pose, targeting for transforming single-view image into low-resolution 3D-aware sequential images (i.e., orbit video with 512×512 resolution). In the second stage, this low-resolution orbit video is further fed into 3D-aware video-to-video refiner with additional depth condition, leading to high-resolution orbit video (1,024×1,024) with highly detailed texture. Considering that the obtained high-resolution orbit video contains a fixed number of multi-view images, we augment them with more novel views through 3D Gaussian Splatting. The resultant dense high-resolution sequential images effectively ease the final 3D reconstruction, yielding high-quality 3D meshes.

The main contribution of this work is the proposal of the two-stage video diffusion based paradigm that fully unleashes the power of inherent 3D prior knowledge in the pre-trained video diffusion model to strengthen image-to-3D generation. This also leads to the elegant views of how video diffusion model should be designed for fully exploiting 3D geometry priors, and how to scale up the resolution of multi-view images for high-resolution image-to-3D generation. Extensive experiments demonstrate the state-of-the-art performances of our Hi3D on both novel view synthesis and single view reconstruction tasks.

## 2 RELATED WORKS

### 2.1 Image-to-3D

Reconstructing 3D models from a single view poses a significant challenge. Early attempts [11, 13, 24, 27, 57] to reconstruct 3D shapes from single images through regression [27] or retrieval methods [57] struggled with generalization to real-world data or new object categories. However, the rapid advancements in generative models, particularly diffusion models [18, 19], have opened up new avenues. These models have demonstrated remarkable capability in generating a broad spectrum of images [35, 40, 48–50] and videos [2, 3, 12, 23, 58, 68], offering fresh perspectives for 3D asset generation using the strong priors inherent in 2D diffusion models. Dreamfusion [44] firstly utilizes powerful text-to-image diffusion models [50] as prior knowledge for text-to-3D generation, showcasing remarkable improvements in 3D outputs. This pioneering work has prompted a series of subsequent studies that further refine and strengthen this method for image-to-3D generation [5, 36, 45, 47, 56, 63]. While these methods have shown promising results, they often require extensive time for textual inversion [47] and optimization of Neural Radiance Fields (NeRF) [37], leading to inefficiencies and multi-faces (Janus) issue. The second category of research explores the direct training of 3D diffusion models across various 3D representations, including point clouds [34, 41, 65], meshes [15, 31], and neural fields [1, 4, 6, 16, 22, 26, 38, 43, 60, 66]. Nonetheless, the limited availability of diverse 3D data has hampered these models' ability to generalize, with many studies being validated only on a narrow range of shape categories.

The third line works involves using diffusion models to generate multi-view images firstly, and reconstructing 3D model from the images. The core of this category of methods lies in generating consistent multi-view images. Zero123 [29] finetunes a stable diffusion model [49] to generate novel views of an object. Zero123-XL [7], Stable-Zero123 [54] improve the ability of Zero123 by improving the training data. However, these approaches generate multi-view images independently, leading to potential inconsistencies among the views. [21, 30, 32, 51, 52, 55] propose to generate multi-views of an object simultaneously. SyncDreamer [30] employs 3D feature volume, Wonder3D [32] and EpiDiff [21] use multi-view attention mechanisms for maintaining multi-view consistency. However, these methods typically produce only low-resolution images, restricted by the significant memory requirements of 3D representations or the computational intensity of attention mechanisms. In our work, we fall into the third group of Image-to-3D generation. We approach multi-view generation as an image-to-video task, and

design a two-stage video diffusion based paradigm that fully unleashes the power of inherent 3D prior knowledge in the pre-trained video diffusion model to strengthen image-to-3D generation.

## 2.2 Reconstruction from Multi-view images

The recent success of neural radiance fields (NeRFs) [37] has inspired many follow-up works [14, 39, 59] to achieve impressive 3D reconstruction. However, these methodologies typically necessitate over a hundred images for training views, and their efficacy in reconstructing 3D models from sparse multi-view images remains suboptimal. To address this issue, several studies have endeavored to minimize the requisite number of training views. For instance, DS-NeRF [9] introduced additional depth supervision to enhance rendering quality, while RegNeRF [42] developed a depth smoothness loss for geometric regularization to facilitate training stability. Sparseneus [33] focused on learning geometry encoding priors from image features for adaptable neural surface learning from sparse input views, though the detail in reconstruction results was still lacking. In our work, we have developed a straightforward yet efficient reconstruction pipeline that leverages the state-of-the-art 3D Gaussian Splatting algorithm [25] to augment the generated multi-view images, which enables us to stably and effectively reconstruct high-quality meshes.

## 3 PRELIMINARIES

**Video Diffusion Models.** Diffusion models [18, 53] are generative models that can learn the target data distribution from a Gaussian distribution through a gradual denoising process. Video diffusion models [3, 20] are usually built upon pre-trained image diffusion models [40, 49], and enable the denoising process over multiple frames simultaneously. For simplicity, we adopt Stable Video Diffusion [2] as the basic video diffusion model, which achieves state-of-the-art performance in image-to-video generation. Formally, given a single frame $x^0$, video diffusion model can generate a high-fidelity video consisting of $N$ sequential frames $\mathbf{x} = \{x^0, x^1, ..., x^{(N-1)}\}$ through an iterative denoising process. Specifically, at each denoising step $t$, video diffusion model predicts the amount of noise added in the sequence through a conditional 3D-UNet $\Phi$, and then denoises the sequence by subtracting the predicted noise:

$$\mathbf{x}_{t-1} = \Phi(\mathbf{x}_t; t, c), \tag{1}$$

where $c$ is the condition embedding of the input frame. In practice, Stable Video Diffusion is built within a latent diffusion framework [49] to reduce computational complexity, i.e., operating diffusion process in an encoded latent space. In this way, the input video sequence is first encoded into a latent code by a pre-trained VAE encoder and the denoised latent code is decoded back to pixel space using a VAE decoder after the denoising steps. Note that Stable Video Diffusion is pre-trained on large-scale high-quality video datasets and demonstrates impressive image-to-video generation capacity. In this work, we propose to inherit the underlying temporal consistency knowledge in video diffusion model to boost the multi-view consistency for image-to-3D generation.

**3D Gaussian Splatting.** 3D Gaussian Splatting (3DGS) [25] emerges as a recent groundbreaking technique for novel view synthesis. Unlike 3D implicit representation methods (e.g., Neural Radiance Fields (NeRF) [37]) that rely on computationally intensive volume rendering for image generation, 3DGS achieves real-time rendering speeds through a splatting approach [64]. Specifically, 3DGS represents a 3D scene as a set of scaled 3D Gaussian primitives, and each scaled 3D Gaussian $G_k$ is parameterized by an opacity (scale) $\alpha_k \in [0, 1]$, view-dependent color $c_k \in \mathbb{R}^3$, center position $\mu_k \in \mathbb{R}^{3 \times 1}$, covariance matrix $\sum_k \in \mathbb{R}^{3 \times 3}$. The 3D Gaussians can be queried as follows:

$$G_k(\boldsymbol{x}) = e^{-\frac{1}{2}(\boldsymbol{x} - \mu_k)^T \sum_k^{-1} (\boldsymbol{x} - \mu_k)}. \tag{2}$$

3DGS computes the color of each pixel via alpha blending according to the primitive's depth order $1, ..., K$:

$$C(\boldsymbol{x}) = \sum_{k=1}^{K} c_k \sigma_k \prod_{j=1}^{k-1} (1 - \sigma_j), \sigma_k = \alpha_k G_k(\boldsymbol{x}). \tag{3}$$

Since the rendering process in 3DGS is fast and differentiable, the parameters of 3D Gaussian can be efficiently optimized through a multi-view loss (see [25] for more details). In this paper, we integrate 3DGS into our 3D reconstruction pipeline to extract high-fidelity meshes, tailored for synthesized high-resolution multi-view images.

## 4 OUR APPROACH

In this work, we devise a new High-resolution image-to-3D generation architecture, namely Hi3D, to novelly integrate video diffusion models into 3D-aware 360° sequential image generation (i.e., orbital video generation). Our launching point is to exploit the intrinsic temporal consistent knowledge in video diffusion models to enhance cross-view consistency in 3D generation. We begin this section by elaborating the problem formulation of image-to-3D generation (Sec. 4.1). We then elaborate the details of two-stage video diffusion based paradigm in our Hi3D framework. Specifically, in the first stage, we remould the pre-trained image-to-video diffusion model with additional condition of camera pose and then fine-tune it on 3D data to enable orbital video generation (Sec. 4.2). In the second stage, we further scale up the multi-view image resolution through a 3D-aware video-to-video refiner (Sec. 4.3). Finally, a novel 3D reconstruction pipeline is introduced to extract high-quality 3D mesh from these high-resolution multi-view images (Sec. 4.4). The whole architecture of Hi3D is illustrated in Figure 2.

## 4.1 Problem Formulation

Given a single RGB image $\mathbf{I} \in \mathbb{R}^{3 \times H \times W}$ (source view) of an object $X$, our target is to generate its corresponding 3D content (i.e., textured triangle mesh). Similar to previous image-to-3D generation methods, we also decompose this challenging task into two steps: 1) generate a sequence of multi-view images around the object $X$ and 2) reconstruct the 3D content from these generated multi-view images. Technically, we first synthesize a sequence of multi-view images $\mathbf{F} \in \mathbb{R}^{N \times 3 \times H \times W}$ of the object from $N$ different camera poses $\boldsymbol{\pi} \in \mathbb{R}^{N \times 3 \times 4}$ corresponding to the input condition image $\mathbf{I}$ in a two-stage manner. Herein, we generate $N = 16$ multi-view images with a high resolution of $H \times W = 1,024 \times 1,024$ around the object in this work. It is worth noting that previous state-of-the-art

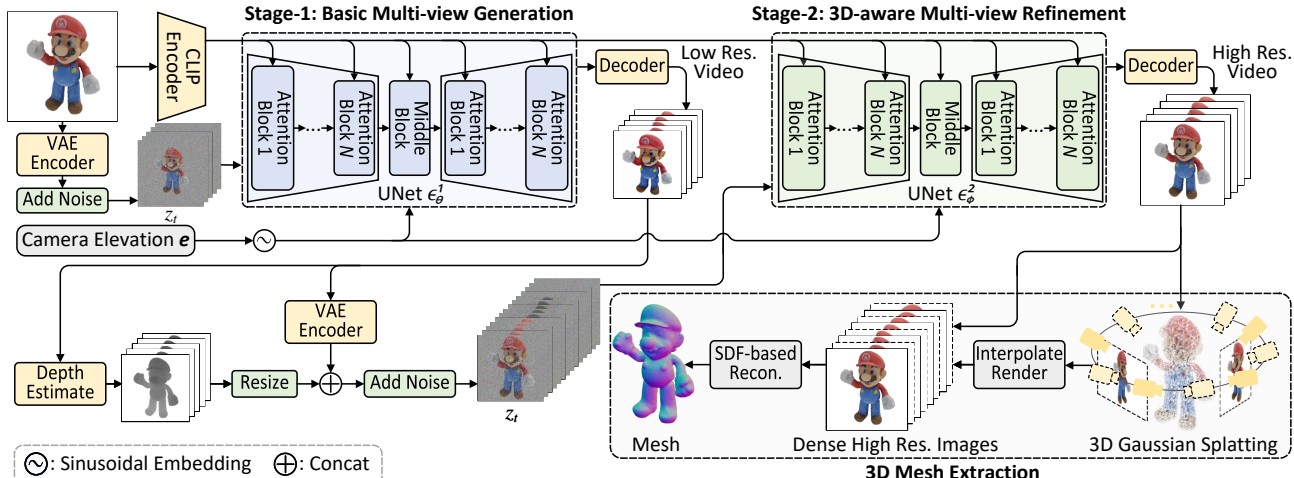

**Figure 2: An overview of our proposed Hi3D. Our Hi3D fully exploits the capabilities of large-scale pre-trained video diffusion models to effectively trigger high-resolution image-to-3D generation. Specifically, in the first stage of basic multi-view generation, Hi3D remoulds video diffusion model with additional camera pose condition, aiming to transform single-view image into low-resolution 3D-aware sequential images. Next, in the second stage of 3D-aware multi-view refinement, we feed this low-resolution orbit video into 3D-aware video-to-video refiner with additional depth condition, leading to high-resolution orbit video with highly detailed texture. Finally, we augment the resultant multi-view images with more novel views through 3D Gaussian Splatting and employ SDF-based reconstruction to extract high-quality 3D meshes.**

image-to-3D models [21, 30, 32] can only generate low-resolution (i.e., $256 \times 256$) multi-view images. In contrast, to the best of our knowledge, our work is the first to enable high-resolution (i.e., $1,024 \times 1,024$) image-to-3D generation, which can preserve richer geometry and texture details of the input image. Next, we extract 3D mesh from these synthesized high-resolution multi-view images through our carefully designed 3D reconstruction pipeline. Since the number of generated views is somewhat limited, it is difficult to extract a high-quality mesh from these sparse views. To alleviate this issue, we leverage the novel view synthesis method (3D Gaussian Splatting [25]) to reconstruct an implicit 3D model from multi-view images $\mathbf{F}$. Then we render additional interpolation views $\mathbf{F}^* \in \mathbb{R}^{M \times 3 \times H \times W}$ between the multi-view images and add these rendered views into $\mathbf{F}$, thereby obtaining dense view images $\mathbf{K} \in \mathbb{R}^{(N+M) \times 3 \times H \times W} = \mathbf{F} + \mathbf{F}^*$ of the object $X$. Finally, we adopt an SDF-based reconstruction method [59] to extract a high-quality mesh from these dense views $\mathbf{K}$.

### 4.2 Stage-1: Basic Multi-view Generation

Previous image-to-3D generation methods [21, 30, 32, 51] usually rely on pre-trained image diffusion models to accomplish multi-view generation. These methods generally extend the 2D UNet in image diffusion models to 3D UNet by injecting multi-view cross-attention layers. These added attention layers are trained from scratch on 3D datasets to learn multi-view consistency. However, the image resolution in these methods is restricted to $256 \times 256$ to ensure training stability. Maintaining the original resolution ($512 \times 512$) in pre-trained image diffusion models will lead to slower convergence and higher variance, as pointed in Zero123 [29]. Consequently, due to such low-resolution limitation, these methods fail

to fully capture the primary rich 3D geometry and texture details in the input 2D image. In addition, we observe that these approaches still suffer from multi-view inconsistency issue, especially for complex object geometry. This may be attributed to the fact that the underlying pre-trained 2D diffusion model is exclusively trained on individual 2D images and lacks 3D modeling of multi-view correlation. To alleviate the above issues, we redefine single image to multi-view images as 3D-aware sequence image generation (i.e., orbital video generation) and utilize pre-trained video diffusion models to fulfill this goal. In particular, we repurpose Stable Video Diffusion (SVD) [2] to generate multi-view images from the input image. SVD is appealing because it was trained on a large variety of videos, which allows the network to encounter multiple views of an object during training. This potentially alleviates the 3D data scarcity problem. Moreover, SVD has already explicitly modeled the multi-frame relation via temporal attention layers. We can inherit the intrinsic multi-frame consistent knowledge in these temporal layers to pursue multi-view consistency in 3D generation.

**Training Data.** We first construct a high-resolution multi-view image dataset from the LVIS subset of the Objaverse [8]. For each 3D asset, we render 16 views with $1,024 \times 1,024$ resolution at random elevation $e \in [-10°, 40°]$. It is important to note that while the elevation is randomly selected, it remains the same across all views within a single video. For each video, the cameras are positioned equidistantly from the object with distance $r = 1.5$ and spaced evenly from $0°$ to $360°$ in azimuth angle. In total, our training dataset comprises approximately $300,000$ videos, denoted as $\mathcal{J} = \{(\mathbf{J}_i, \mathbf{I}_i, e_i)\}$, where the input condition image $\mathbf{I}_i = [\mathbf{J}_i]_1$ is the first frame in sequential images $\mathbf{J}_i$.

**Video Diffusion Fine-tuning.** In the first stage, our goal is to repurpose the pre-trained image-to-video diffusion model to

generate multi-view consistent sequential images. The aforementioned multi-view image dataset $\mathcal{J} = \{(\mathbf{J}_i, \mathbf{I}_i, e_i)\}$ is thus leveraged to fine-tune the 3D-aware video diffusion model with additional camera pose condition. Specifically, given the input single-view image $\mathbf{I_i}$, we first project it into latent space by the VAE encoder of video diffusion model, and channel-wisely concatenate it with the noisy latent sequence, which encourages synthesized multi-view images to preserve the identity and intricate details of the input image. In addition, we incorporate the input condition image's CLIP embeddings [46] into the diffusion UNet through cross-attention mechanism. Within each transformer block, the CLIP embedding matrix acts as the key and value for the cross-attention layers, coupled with the layer's features serving as the query. In this way, the high-level semantic information of the input image is propagated into the video diffusion model. Since the multi-view image sequence is rendered at random elevations, we send the elevation parameter into the video diffusion model as additional condition. Most specifically, the camera elevation angle $e$ is first embedded into sinusoidal positional embeddings and then fed into the UNet along with the diffusion noise timestep $t$. As all multi-view sequences follow the same azimuth trajectory, we do not send the azimuth parameter into the diffusion model. Herein, we omit the original "fps id" and "motion bucket id" conditions in video diffusion model as these conditions are irrelevant to multi-view image generation.

In general, the denoising neural network (3D UNet) in our remolded video diffusion model can be represented as $\epsilon_\theta^1(\mathbf{z}_t; \mathbf{I}, t, e)$. Given the multi-view image sequence $\mathbf{J}$, the pre-trained VAE encoder $\mathcal{E}(\cdot)$ first extracts the latent code of each image to constitute a latent code sequence $\mathbf{z}$. Next, Gaussian noise $\epsilon \sim N(0, I)$ is added to $\mathbf{z}$ through a typical forward diffusion procedure at each time step $t$ to get the noise latent code $\mathbf{z}_t$. The 3D UNet $\epsilon_\theta^1(\mathbf{z}_t; \mathbf{I}, t, e)$ with parameter $\theta$ is trained to estimate the added noise $\epsilon$ based on the noisy latent code $\mathbf{z}_t$, input image condition $\mathbf{I}$ and elevation angle $e$ through the standard mean square error (MSE) loss:

$$\mathcal{L}_{Stage-1} = \mathbb{E}_{\mathbf{I}, \mathbf{J}, e, t, \epsilon} \left[ ||w(t)(\epsilon_\theta^1(\mathbf{z}_t; \mathbf{I}, e, t) - \epsilon)||_2^2 \right], \quad (4)$$

where $w(t)$ is a corresponding weighing factor.

Instead of directly training denoising neural network in high resolution (i.e., $1,024 \times 1,024$), we decompose this non-trivial problem into more stable sub-problems in a coarse-to-fine manner. In the first stage, we train the denoising neural network by using Eq. (4) with $512 \times 512$ resolution for low-resolution multi-view image generation. The second stage further transforms $512 \times 512$ multi-view images into high-resolution ($1,024 \times 1,024$) multi-view images.

## 4.3 Stage-2: 3D-aware Multi-view Refinement

The output $512 \times 512$ multi-view images of Stage-1 exhibit promising multi-view consistency, while still failing to fully capture the geometry and texture details of inputs. To address this issue, we include an additional stage to further scale up the low-resolution outputs of the first stage through a new 3D-aware video-to-video refiner, leading to higher-resolution (i.e., $1,024 \times 1,024$) multi-view images with finer 3D details and consistency.

In this stage, we also remould the pre-trained video diffusion model as 3D-aware video-to-video refiner. Formally, such denoising neural network can be formulated as $\epsilon_\phi^2(\mathbf{z}_t; \mathbf{I}, \hat{\mathbf{J}}, \mathbf{D}, t, e)$, where $\hat{\mathbf{J}}$

denotes the generated multi-view images corresponding the input image $\mathbf{I}$ in Stage-1, $\mathbf{D}$ is the estimated depth sequence of the generated multi-view images $\hat{\mathbf{J}}$. To be clear, the input conditions $\mathbf{I}$ and $e$ are injected into pre-trained video diffusion model by the same way as in Stage-1. Besides, we adopt the VAE encoder to extract the latent code sequence of the pre-generated multi-view images $\hat{\mathbf{J}}$ and channel-wisely concatenate them with the noise latent $\mathbf{z}_t$ as conditions. Moreover, to fully exploit the underlying geometry information of the generated multi-view images, we leverage an off-the-shelf depth estimation model to estimate the depth of each image in $\hat{\mathbf{J}}$ as 3D cues, yielding a depth map sequence $\mathbf{D}$. We then directly resize the depth maps into the same resolution of the latent code $\mathbf{z}_t$, and channel-wisely concatenate them with $\mathbf{z}_t$. Finally, the remoulded denoising neural network is trained through standard MSE loss in diffusion models:

$$\mathcal{L}_{Stage-2} = \mathbb{E}_{\mathbf{I}, \mathbf{J}, \hat{\mathbf{J}}, \mathbf{D}, e, t, \epsilon} \left[ ||w(t)(\epsilon_\phi^2(\mathbf{z}_t; \mathbf{I}, \hat{\mathbf{J}}, \mathbf{D}, e, t) - \epsilon)||_2^2 \right], \quad (5)$$

where $w(t)$ is a weighing factor. Note that the resolution of training images in Eq. (5) is scaled up to $1,024 \times 1,024$.

During training, we adopt some image degradation methods [61] to synthesize $\hat{\mathbf{J}}$ for data augmentation, instead of solely using the generated coarse multi-view images from Stage-1. In particular, we utilize a high-order degradation model to synthesize training data, including a series of blur, resize, noise, and compression processes. To replicate overshoot artifacts (e.g., ringing or ghosting around sharp transitions in images), we utilize *sinc* filter. Additionally, random masking techniques are used to simulate the effect of shape deformation. This way not only accelerates the training process, but also enhances the robustness of our video-to-video refiner.

## 4.4 3D Mesh Extraction

Through the above two-stage video diffusion based paradigm, we can obtain a high-resolution image sequence $\mathbf{F} \in \mathbf{R}^{N \times 3 \times H \times W}$ ($N = 16, H = W = 1,024$) conditioned on the input image $\mathbf{I}$. In this section, we aim to extract high-quality meshes from these generated high-resolution multi-view images. Previous image-to-3D methods [21, 30, 32] usually reconstruct the target 3D mesh from the output image sequence by optimizing the neural implicit Signed Distance Field (SDF) [17, 59]. Nevertheless, these SDF-based reconstruction methods are originally tailored for dense image sequences captured in the real world, which commonly fail to reconstruct high-quality mesh based on only sparse views.

To alleviate this issue, we design a unique 3D reconstruction pipeline for high-resolution sparse views. Instead of directly adopting SDF-based reconstruction methods to extract 3D mesh, we first use the 3D Gaussian Splatting (3DGS) algorithm [25] to learn an implicit 3D model from the generated high-resolution image sequence. 3DGS has demonstrated remarkable novel view synthesis capabilities and impressive rendering speed. Herein we attempt to utilize 3DGS's implicit reconstruction ability to augment the output sparse multi-view images of Stage-2 with more novel views. Specifically, we render $M$ interpolation views $\mathbf{F}^*$ between the adjacent images in $\mathbf{F}$ from the reconstructed 3DGS. Finally, we optimize an SDF-based reconstruction method [59] based on the augmented dense views $\mathbf{F} + \mathbf{F}^*$ to extract the high-quality 3D mesh of the object $X$.

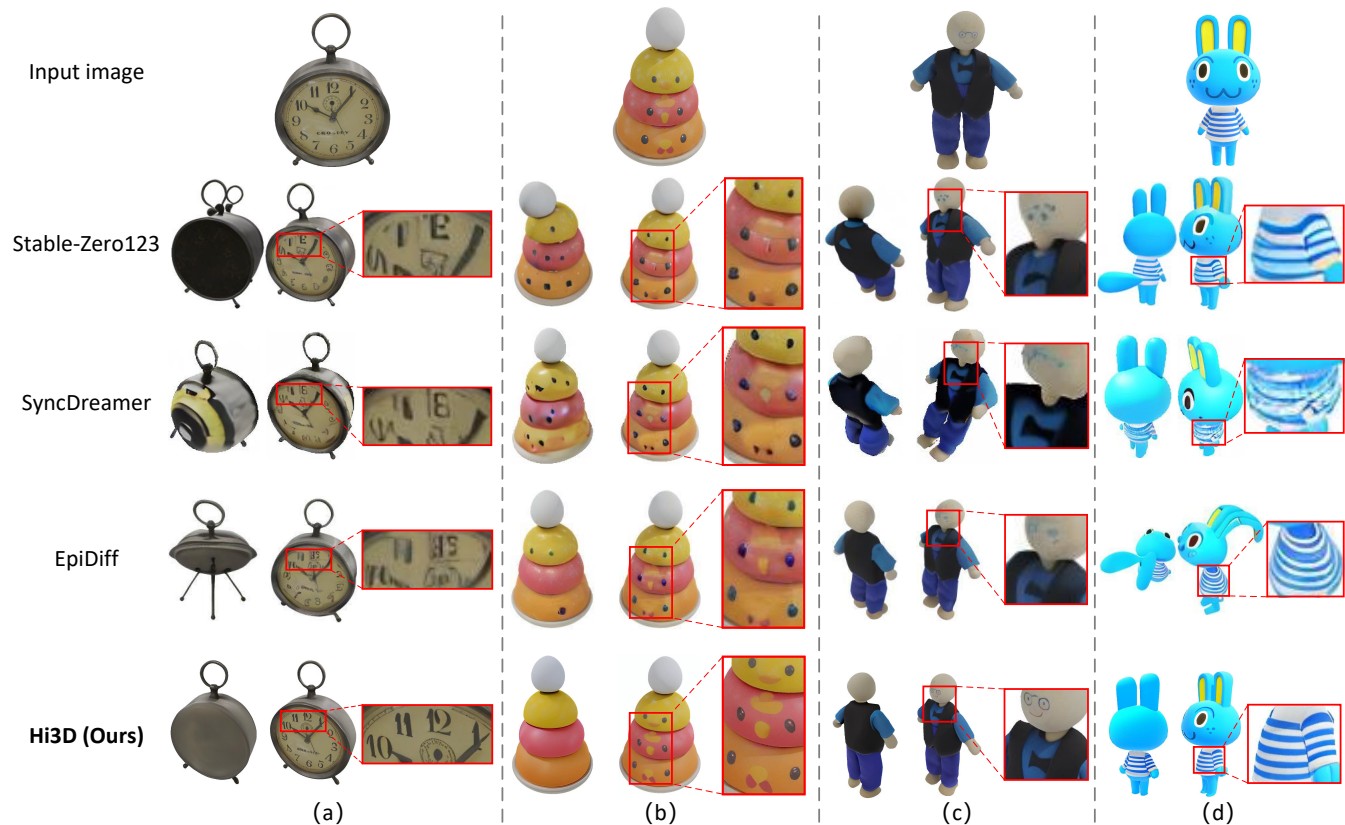

**Figure 3: Qualitative comparisons with Stable-Zero123 [54], SyncDreamer [30] and EpiDiff [21] on novel view synthesis task. Our Hi3D generates high-resolution multi-view images with remarkable consistent details.**

**Table 1: Quantitative comparison with state-of-the-art methods in novel view synthesis on GSO dataset.**

| Method | PSNR↑ | SSIM↑ | LPIPS↓ |
|---|---|---|---|
| Realfusion [36] | 15.26 | 0.722 | 0.283 |
| Zero123 [29] | 18.93 | 0.779 | 0.166 |
| Zero123-XL [7] | 19.47 | 0.783 | 0.159 |
| Stable-Zero123 [54] | 19.79 | 0.788 | 0.153 |
| SyncDreamer [30] | 20.05 | 0.798 | 0.146 |
| EpiDiff [21] | 20.49 | 0.855 | 0.128 |
| **Hi3D (Ours)** | **24.26** | **0.864** | **0.119** |

## 5 EXPERIMENTS

### 5.1 Experimental Settings

**Datasets and Evaluation.** We empirically validate the merit of our Hi3D model by conducting experiments on two primary tasks, i.e., novel view synthesis and single view reconstruction. Following [21, 30, 32], we perform quantitative evaluation on Google Scanned Object (GSO) dataset [10]. For novel view synthesis task, we employ three commonly adopted metrics: PSNR, SSIM [62], and LPIPS [67]. For the single view reconstruction task, we use Chamfer Distances and Volume IoU to measure the quality of the reconstructed 3D models. In addition, to assess the generalization ability of our Hi3D,

we perform qualitative evaluation over single images with various styles derived from the internet.

**Implementation Details.** During the first stage of basic multi-view generation, we downscale the video dataset as $512 \times 512$ videos. For the second stage of multi-view refinement, we not only feed the outputs of the first stage, but also adopt synthetic data generation strategy (similar to traditional image/video restoration methods [61]) for data augmentation. This strategy aims to accelerate the training process and enhance the model's robustness. The overall experiments are conducted on eight 80G A100 GPUs. Specifically, the first stage undergoes $80,000$ training steps (approximately 3 days), with a learning rate of $1 \times 10^{-5}$ and a total batch size of 16. The second stage contains $20,000$ training steps (around 3 days), with a learning rate of $5 \times 10^{-5}$ and a reduced batch size of 8.

**Compared Methods.** We compare our Hi3D with the following state-of-the-art methods: RealFusion [36] and Magic123 [45] exploit 2D diffusion model (Stable Diffusion [49]) and SDS loss [44] for reconstructing from single-view image. Zero123 [29] learns to generate novel view images of the same object from different viewpoints, and can be integrated with SDS loss for 3D reconstruction. Zero123-XL [7] and Stable-Zero123 [54] further upgrade Zero123 by enhancing the training data quality. One-2-3-45 [28] directly learns explicit 3D representation via 3D Signed Distance Functions (SDFs) [33] from multi-view images (i.e., the outputs

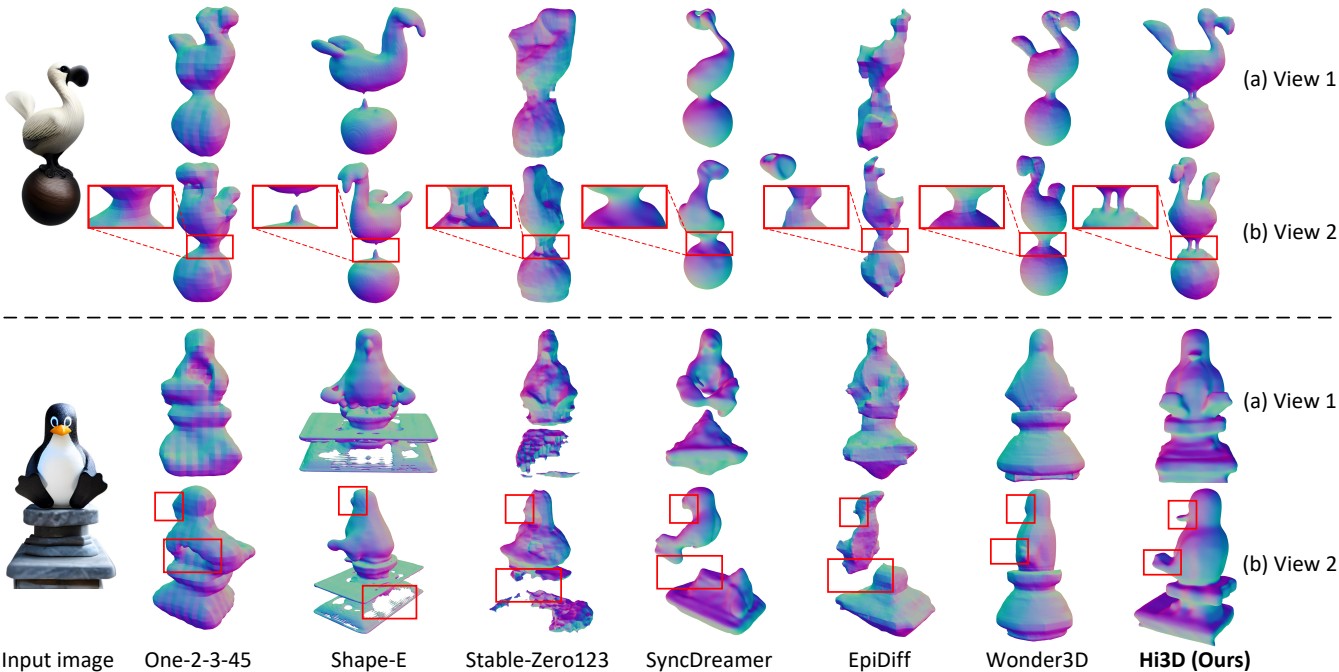

Figure 4: Qualitative comparison of 3D meshes generated by various methods on single view reconstruction task.

of Zero123). Point-E [41] and Shap-E [22] are pre-trained over an extensive internal OpenAI 3D dataset, thereby being capable of directly transforming single-view images into 3D point clouds or shapes encoded in MLPs. SyncDreamer [30] introduces a 3D global feature volume to maintain multi-view consistency. Wonder3D [32] and EpiDiff [21] leverage 3D attention mechanisms to enable inter-action among multi-view images via cross-attention layers. Note that in novel view synthesis task, we only include partial baselines (i.e., Zero123 series, SyncDreamer, EpiDiff) that can produce exactly the same viewpoints as our Hi3D for fair comparison.

## 5.2 Novel View Synthesis

Table 1 summarizes performance comparison on novel view synthe-sis task, and Figure 3 showcases qualitative results in two different views. Overall, our Hi3D consistently exhibits better performances than existing 2D diffusion based approaches. Specifically, Hi3D achieves the PSNR of 24.26%, which outperforms the best competi-tor EpiDiff by 3.77%. The highest image quality score of our Hi3D generally highlights the key advantage of video diffusion based paradigm that exploits 3D prior knowledge to boost novel view synthesis. In particular, due to the independent image translation, Zero123 series (e.g., Stable-Zero123) fails to achieve multi-view con-sistency results (e.g., one/two rings on the head of the alarm clock in different views in Figure 3 (a)). SyncDreamer and EpiDiff further strengthen multi-view consistency by exploiting 3D intermediate in-formation or using multi-view attention mechanisms. Nevertheless, their novel-view results still suffer from blurry and unrealistic is-sues with degraded image quality (e.g., the blurry numbers of alarm clock in Figure 3 (a)) due to the restricted low image resolution (256×256). Instead, by mining 3D priors and scaling up multi-view image resolution via video diffusion model, our Hi3D manages

Table 2: Quantitative comparison with state-of-the-art meth-ods in single view reconstruction on GSO dataset.

| Method | Chamfer Dist.↓ | Volume IoU↑ |
|---|---|---|
| Realfusion [36] | 0.0819 | 0.2741 |
| Magic123 [45] | 0.0516 | 0.4528 |
| One-2-3-45 [28] | 0.0629 | 0.4086 |
| Point-E [41] | 0.0426 | 0.2875 |
| Shap-E [22] | 0.0436 | 0.3584 |
| Stable-Zero123 [54] | 0.0321 | 0.5207 |
| SyncDreamer [30] | 0.0261 | 0.5421 |
| EpiDiff [21] | 0.0343 | 0.4927 |
| Wonder3D [32] | 0.0199 | 0.6244 |
| **Hi3D (Ours)** | **0.0172** | **0.6631** |

to produce multi-view consistent and high-resolution 1,024×1,024 images, leading to highest image quality (e.g., the clearly visible numbers in alarm clock in Figure 3 (a)).

## 5.3 Single View Reconstruction

Next, we evaluate the single view reconstruction performance of our Hi3D in Table 2. In addition, Figure 4 shows qualitative comparison between Hi3D and existing methods. In general, our Hi3D outper-forms state-of-the-art methods over both two metrics. Specifically, One-2-3-45 directly leverages multi-view outputs of Zero123 with sub-optimal 3D consistency for reconstruction, which commonly results in over-smooth meshes with fewer details. Stable-Zero123 further improves 3D consistency with higher-quality training data, while still suffering from missing or over-smooth meshes. Differ-ent from independent image translation in Zero123, SyncDreamer, EpiDiff, and Wonder3D exploit simultaneous multi-view image

Anonymous Authors

**Table 3: Ablation study on 3D-aware multi-view refinement.**

| Setting | PSNR↑ | SSIM↑ | LPIPS↓ |
|---|---|---|---|
| Hi3D | **24.26** | **0.864** | **0.119** |
| w/o refinement | 22.09 | 0.842 | 0.136 |
| w/o depth | 23.12 | 0.848 | 0.128 |

**Table 4: Ablation study on 3D reconstruction pipeline.**

| Setting | Chamfer Dist.↓ | Volume IoU↑ |
|---|---|---|
| $M = 0$ | 0.0186 | 0.6375 |
| $M = 16$ | **0.0172** | **0.6631** |
| $M = 32$ | 0.0174 | 0.6598 |
| $M = 48$ | 0.0175 | 0.6607 |

translation through 2D diffusion model, thereby leading to better 3D consistency. However, they struggle to reconstruct complex 3D meshes with rich details due to the limitation of low-resolution multi-view images. In contrast, our Hi3D fully unleashes the power of inherent 3D prior knowledge in pre-trained video diffusion model and scales up the multi-view images into higher resolution. Such design enables higher-quality 3D mesh reconstruction with richer fine-grained details (e.g., the feet of bird and penguin in Figure 4).

## 5.4 Ablation Studies

**Effect of 3D-aware Multi-view Refinement Stage.** Here we examine the effectiveness of the second stage (i.e., 3D-aware multi-view refinement) on novel view synthesis. Table 3 details the performances of ablated runs of our Hi3D. Specifically, the second row removes the whole second stage, and the performances drop by a large margin. This validates the merit of scaling up multi-view image resolution via 3D-aware video-to-video refiner. In addition, when only removing depth condition in second stage (row 3), a clear performance drop is attained, which demonstrates the effectiveness of depth condition that enhances 3D geometry consistency among multi-view images.

**Effect of Interpolation view number $M$ in 3D Reconstruction.** Table 4 shows the single view reconstruction performances of using different numbers of interpolation views $M$. In the extreme case of $M = 0$, no interpolation view is employed, and the 3D reconstruction pipeline degenerates to typical SDF-based reconstruction. By increasing $M$ as 16, the reconstruction performances are clearly improved, which basically shows the advantage of interpolation views via 3DGS. However, when further enlarging $M$, the performances slightly decrease. We speculate that this may be the result of unnecessary information across views repeat and error accumulating. In practice, $M$ is generally set to 16.

## 5.5 More Discussions

**Text-to-image-to-3D.** By integrating advanced text-to-image models (e.g., Stable Diffusion [49], Imagen [50]) into our Hi3D, we are capable of generating 3D models directly from textual descriptions, as illustrated in Figure 5. Our approach manages to produce higher-fidelity 3D models with highly-detailed texture, which again highlights the merit of high-resolution multi-view image generation with 3D consistency.

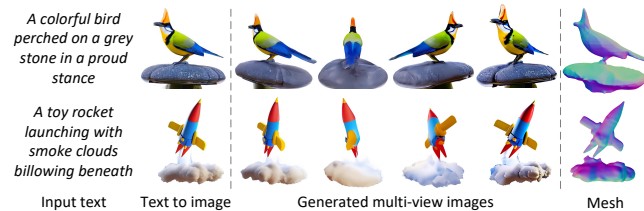

**Figure 5: Examples of using Hi3D for text-to-3D generation.**

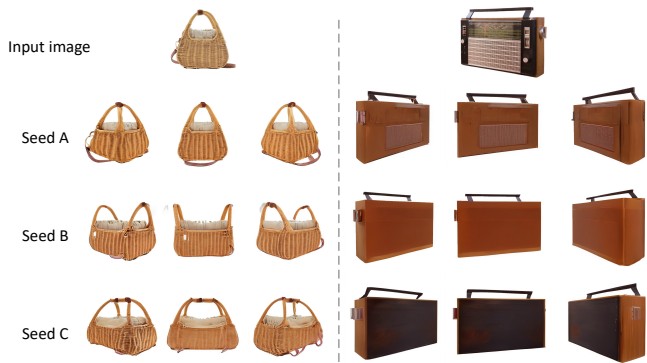

**Figure 6: Diverse and creative results of our Hi3D with different seeds.**

**Diversity and Creativity in 3D Model Generation.** Here we examine the diversity and creativity of our Hi3D by using different random seeds. As shown in Figure 6, our Hi3D is able to generate diverse and plausible instances, each with distinct geometric structures or textures. This capability not only enhances the flexibility of 3D model creation but also significantly contributes to the exploration of creative possibilities in 3D design and visualization.

## 6 CONCLUSION

This paper explores inherent 3D prior knowledge in pre-trained video diffusion model for boosting image-to-3D generation. Particularly, we study the problem from a novel viewpoint of formulating single image to multi-view images as 3D-aware sequential image generation (i.e., orbital video generation). To materialize our idea, we have introduced Hi3D, which executes two-stage video diffusion based paradigm to trigger high-resolution image-to-3D generation. Technically, in the first stage of basic multi-view generation, a video diffusion model is remoulded with additional 3D condition of camera pose, targeting for transforming single image into low-resolution orbital video. In the second stage of 3D-aware multi-view refinement, a video-to-video refiner with depth condition is designed to scale up the low-resolution orbital video into high-resolution sequential images with rich texture details. The resulting high-resolution outputs are further augmented with interpolation views through 3D Gaussian Splatting, and SDF-based reconstruction is finally employed to achieve 3D meshes. Experiments conducted on both novel view synthesis and single view reconstruction tasks validate the superiority of our proposal over state-of-the-art approaches.

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
