# OpenReview forum: "Hi3D: Pursuing High-Resolution Image-to-3D Generation with Video Diffusion Models"
_acmmm.org/ACMMM/2024/Conference — MM2024 Poster_

### Official Review · Reviewer_WuFv · 2024-05-20

**Rating:** 5
**Confidence:** 3

**Summary:**

This article proposes a novel method for synthesizing views with muti-view consistent and high-resolution. It adopts a two-stage coarse to fine generation method. The first stage is the generated paradigm, which expands the visual boundaries with fewer samples and increases the resolution of the image, aiming to generate multi-view related samples. The second stage utilizes the multiple samples generated in the first stage, while combining the advantages of novel 3DGS and SDF scene representations, to further accurately depict the mesh representation of 3D space.

**Strengths:**

(1) The visualization results and videos demonstrate that the proposed method is beneficial for maintaining geometric consistency and synthesizing high-quality high-resolution views.

(2) This experiment shows obvious improvement compared with static methods.

**Limitations:**

(1) Line 574 3DGS is an explicit reconstruction rather than an implicit reconstruction. The article needs to be carefully reviewed and the wording should be consistent.

(2) The citation of the method section should be reasonable. E.g. Is the deep estimation network a monocular or multi-view model?  ( line 352)  This should be explained clearly. If it is an existing method, a reference should be added.

(3) Since the framework consists of many modules, it would be better to analyze the parameters of them, so that we can know how to further improve the efficiency of the method.

(4) The reference format appears to be incorrect and should follow the standards of MM2024.

**Suitability:**

3

---

### Official Review · Reviewer_ZhWA · 2024-05-21

**Rating:** 2
**Confidence:** 3

**Summary:**

This paper proposes using a pre-trained video diffusion model to generate multi-view images for reconstruction. Additionally, a 3D-aware video-to-video refiner is trained to enhance the multi-view images to higher resolutions. Both models are trained in two stages on the Objaverse dataset using the A100 GPU for three days.

**Strengths:**

1. Experiments validate the effectiveness of the architecture, including the 3D-aware Multi-view Refinement stage and the interpolation view number 𝑀 in 3D reconstruction.

2. Hi3D can generate diverse and plausible instances, each with distinct geometric structures or textures, using a video diffusion model.

**Limitations:**

1. Lack of novelty in this approach, as many similar works, such as VFusion3D, V3D, SV3D,  have already been conducted using video models for 3D generation.

2. The pipeline is overly redundant, requiring three stages: Multi-view Generation, 3D-aware Multi-view Refinement, and 3D Mesh Extraction, which includes Gaussian splatting and SDF-based reconstruction. The inference efficiency is too low.

3. There is a lack of experimentation, making the results unconvincing. Additionally, there is no comparison with other video model-based methods, such as VFusion3D, and V3D.

**Suitability:**

3

---

### Official Review · Reviewer_gNKq · 2024-05-25

**Rating:** 6
**Confidence:** 2

**Summary:**

This work is mainly an image-conditioned 3D reconstruction task. The core idea is to carefully finetune the 1.	Higher-resolution sequential image generation is impressive, which is higher than the sota work [1], su3d with a resolution of 576x576.
[1] SV3D: Novel Multi-view Synthesis and 3D Generation from a Single Image using Latent Video Diffusion
video diffusion model to obtain consistent orbital video and then reconstruct high-resolution 3D models in a coarse-to-fine manner.

**Strengths:**

Higher-resolution sequential image generation is impressive, which is higher than the sota work [1], su3d with a resolution of 576x576.
[1] SV3D: Novel Multi-view Synthesis and 3D Generation from a Single Image using Latent Video Diffusion

**Limitations:**

Overall, I have no major concerns. Considering the rapid development of this field, it would be better to discuss the differences and compare with some recent work su3d [1] and InstantMesh [2].
[2] InstantMesh: Efficient 3D Mesh Generation from a Single Image with Sparse-view Large Reconstruction Models

**Suitability:**

3

---

### Official Review · Reviewer_mYVA · 2024-05-25

**Rating:** 4
**Confidence:** 3

**Summary:**

The paper introduces Hi3D, a high-resolution image-to-3D generation framework that leverages video diffusion models. Hi3D aims to address the limitations of current methods in producing multi-view consistent images with high-resolution textures by utilizing a video diffusion paradigm. This approach translates a single image into a series of sequential, multi-view images and ultimately reconstructs high-fidelity 3D meshes.

**Strengths:**

+ Innovative Use of Video Diffusion Models: Hi3D leverages the inherent temporal consistency knowledge in video diffusion models to enhance multi-view consistency in 3D generation, marking a significant improvement over traditional 2D-based methods.

+ Two-Stage Refinement Process: The two-stage approach, which involves generating low-resolution multi-view images and refining them to high resolution, ensures detailed and consistent texture generation. This methodology effectively addresses the limitations of existing low-resolution image-to-3D methods.

+ Comprehensive Evaluation and High-Quality Results: Extensive experiments demonstrate Hi3D's superiority in both novel view synthesis and single-view reconstruction. The method's ability to produce high-resolution, multi-view consistent images with detailed textures is well-validated through qualitative and quantitative comparisons.

**Limitations:**

- Scalability Issues: While Hi3D shows impressive results with the provided dataset, scaling this approach to even higher resolutions or more complex scenes might be challenging.
- The current comparison methods seem to be insufficient, more related works are suggested to be compared. For example, "DreamGaussian: Generative Gaussian Splatting for Efficient 3D Content Creation".

**Suitability:**

3

---

### Meta-Review · Area_Chair_UQE8 · 2024-06-28

**Recommendation:** Accept (Poster)
**Confidence:** 4

**Metareview:**

This paper received three positive reviews and one negative review. The concerns raised in the negative review most focused on the complex pipeline and the incremental novelty. A complex pipeline can be justified by the strong results acknowledged in the other reviews. The concern with the incremental novelty was related to some concurrent work in arxiv. Per ACM MM conference policy, these concurrent papers should not affect the review of this paper. Therefore, we would like to recommend the acceptance of this paper.